# Sleep disturbance as a moderator of the association between physical activity and later pain onset among American adults aged 50 and over: evidence from the Health and Retirement Study

Daniel Whibley [1,2,3] Heidi M Guyer,[4,5] Leslie M Swanson,[6] Tiffany J Braley,[7,8] Anna L Kratz,[2] Galit Levi Dunietz[8]

For numbered affiliations see end of article.

**Correspondence to**
Dr Daniel Whibley;
daniel.whibley@abdn.ac.uk

## ABSTRACT

**Objective** To examine whether sleep disturbance modifies the association between physical activity and incident pain.

**Design** Prospective population-based study.

**Setting** Health and Retirement Study.

**Participants** American adults aged ≥50 years who reported no troublesome pain in 2014 were re-assessed for pain in 2016. Of 9828 eligible baseline respondents, 8036 (82%) had complete follow-up data for adjusted analyses (weighted analysis population N=42 407 222).

**Exposures** Physical activity was assessed via interview with questions about time spent in moderate and vigorous physical activity. Sleep disturbance, assessed using a modified form of the Jenkins Sleep Scale, was examined as a potential moderator.

**Main outcome measure** Troublesome pain.

**Results** In weighted analyses, 37.9% of the 2014 baseline pain-free sample participated in moderate or vigorous physical activity once a week or less, with an overall mean Physical Activity Index Score of 9.0 (SE=0.12). 18.6% went on to report troublesome pain in 2016. Each one-point higher on the Physical Activity Index Score was associated with a reduced odds ratio (OR) of incident pain for those who endorsed sleep disturbance never/rarely (OR=0.97, 95% CI 0.94 to 0.99), but not for those who endorsed sleep disturbance sometimes (OR=0.99, 95% CI 0.97 to 1.01) or most of the time (OR=1.01, 95% CI 0.99 to 1.03). The analysis of possible interaction demonstrated that frequency of sleep disturbance moderated the physical activity and incident pain association (Wald test: p=0.02).

**Conclusions** The beneficial association of physical activity on reduced likelihood of later pain was only observed in persons who endorsed low levels of sleep disturbance.

## INTRODUCTION

Pain is a pre-eminent public health issue,[1] affecting approximately 20% of the global adult population.[2] Its reach is pervasive, impacting on daily functioning,[3] mental

---

### Strengths and limitations of this study

► This study uses data representative of the US population aged 50 years and over to provide evidence for the moderating effect of sleep disturbance on the association between the level of physical activity and later pain onset.

► Strengths of the study include its complex survey design, which supports population-level inference, and the low level of missing data.

► Limitations of the study include the self-reported nature of the variables under study, and the lack of available information on the presence of specific sleep disorders.

---

health[4] and quality of life.[5] The societal burden of pain is also vast, with annual economic costs, including those associated with healthcare and social services, exceeding $560 billion in the USA alone.[6] Half of the older adult population in the USA is estimated to experience troublesome pain.[7] With an increasingly older population[8] and observed associations between pain onset and a reduction in healthy ageing,[9] discovery of strategies to prevent or reduce the proliferation of pain in older adults to support healthy ageing is imperative. The identification of modifiable and protective lifestyle factors that contribute to pain is key to guide population-level efforts to achieve this aim.

One such modifiable factor, and an important determinant of healthy ageing, is physical activity.[10] Physical activity is associated with lower levels of functional disability[11] and morbidity,[12] better mental health[13–15] and a reduced likelihood of pain reporting.[16 17] Accordingly, physical activity is a principal health behaviour included in the WHO's 2015 World Report on Ageing and Health.[18]

BMJ

Like low levels of physical activity, poor sleep quality and short sleep duration are associated with more functional disability,[19] a wide range of diseases,[20–23] poor mental health[24] and pain reporting.[25–28] Poor sleep quality and lower sleep efficiency are also associated with lower levels of physical activity and a decreased likelihood of uptake or maintenance of routine exercise.[29] Conversely, higher levels of physical activity are associated with improvements in subjectively and objectively measured sleep.[30 31] Despite this, advice about sleep behaviour is rarely included in physical activity guidance for older adults. A possible reason for this may be a tacit societal agreement that parameters of sleep (eg, duration, timing and continuity) inevitably decline with age. However, although changes in sleep duration, timing and continuity are observed as humans age, most changes occur during the transition from young to middle adulthood.[32] Indeed, the National Sleep Foundation recommends that older adults sleep 7–8 hours per night (the recommendation for adults aged 18–64 is 7–9 hours).[33]

This study is predicated on the hypothesis that healthy sleep contributes to healthy ageing, and that sleep is an essential component, not only in its own right, but also as a condition within which the benefits of physical activity can be optimally reaped. To investigate the joint impact of sleep and physical activity on health, we examined whether sleep disturbance modified the association between the level of physical activity and likelihood of incident troublesome pain 2 years later in a nationally representative sample of older adults in the USA.

## METHODS
### Study design
We used data from the Health and Retirement Study (HRS), a prospective survey of US adults aged 50 years and older conducted by the University of Michigan. Household face-to-face or telephone interviews are conducted with participants on a biannual basis across the contiguous USA.[34] Data collection is conducted by trained interviewers using laptop computers. The complex survey design of HRS incorporates multistage area probability sampling, stratification, clustering and oversampling of adults of African American and Hispanic race/ethnicity. This allows HRS to provide a nationally representative sample that, with appropriate design-based analyses, can be used to make population-level inferences. A dataset comprising two HRS waves (2014 and 2016) was compiled and used to investigate relationships between physical activity, sleep disturbance and incident pain (those with no pain in 2014 followed forward for pain status in 2016). The HRS is sponsored by the National Institute on Aging.

## MEASURES
### Pain
The outcome was incident troublesome pain, determined by a 'yes' response to the question, 'Are you often troubled with pain?', in 2016. This item has previously been shown to be comparable to pain that at least moderately interferes with daily life.[35]

### Physical activity
The WHO recommends that adults aged ≥65 years participate in 150 min of moderate physical activity or 75 min of vigorous physical activity each week, or an equivalent combination of both.[36] The physical health section of HRS surveys includes questions about current levels of moderate and vigorous physical activity (moderate: 'how often do you take part in sports or activities that are moderately energetic, such as gardening, cleaning the car, walking at a moderate pace, dancing, floor or stretching exercises?'; vigorous: 'How often do you take part in sports or activities that are vigorous, such as running or jogging, swimming, cycling, aerobics or gym workout, tennis, or digging with a spade or shovel?'). Response options available for both questions are hardly ever or never, one to three times a month, once a week or more than once a week. Responses were used to create a Physical Activity Index Score, calculated following a previously used method.[37 38] Moderate activity responses were coded as: 0=hardly ever, 1=one to three times a month, 3=once a week and 6=more than once a week. Vigorous activity responses were coded as: 0=hardly ever, 2=one to three times a month, 6=once a week and 12=more than once a week. A total Physical Activity Index Score was calculated by summing scored responses to these two questions (possible range: 0–18). To provide context for Physical Activity Index Scores, a binary variable was also created, guided by WHO's physical activity recommendations. Respondents were categorised as participating in moderate or vigorous activity more than once a week, or not. According to this approach, Physical Activity Index Scores lower than 6 were indicative of physical inactivity; scores greater than 9 suggested meeting or exceeding WHO's physical activity recommendations.

### Sleep disturbance
Sleep disturbances assessed at baseline (2014) were investigated as a potential moderator of the physical activity (2014) and later pain reporting (2016) association. Four questions about sleep disturbance type and frequency, adapted from the Jenkins Sleep Scale,[39] were included in the physical health section of HRS: 'How often do you have trouble falling asleep?' 'How often do you have trouble with waking up during the night?' 'How often do you have trouble with waking up too early and not being able to fall asleep again?' 'How often do you feel really rested when you wake up in the morning?' Response options were: rarely or never; sometimes; or most of the time (reverse coded for the question about feeling rested on awakening to ensure consistency with other questions). Respondents were categorised as never or rarely experiencing sleep disturbance if they answered 'rarely or never' to all questions; sometimes, if they answered 'sometimes' but not 'most of the time' to any of these

questions, and most of the time if they answered 'most of the time' to any of these questions.

## Demographic and health characteristics

Age, gender, race/ethnicity and number of years of school education were included as demographic covariates. Five health-related covariates were also adjusted for: body mass index (BMI), self-reported history of depression ('Has a doctor ever told you that you have had problems with depression?'), major disease (self-reported history of cancer (any kind except skin), lung disease, heart condition or stroke), diabetes and arthritis.

## STATISTICAL ANALYSIS

Analyses were conducted using Stata V.15.1 (StataCorp LLC, College Station, Texas, USA). HRS sample weights were used to correct for unequal probability of selection that may have arisen as a result of the multistage sampling design. Demographic and health-related characteristics were summarised (mean and SE). To investigate whether sleep disturbance reported in 2014 moderated the relationship between level of physical activity at this time and incident pain in 2016, logistic regression was performed with a Physical Activity Index Score–sleep disturbance status interaction term. As the research question focused on those with no troublesome pain at baseline, subpopulation estimation was used to compute point and variance estimates.[40] This ensured that those reporting no pain at baseline contributed data to the calculation of effect estimates while those with and without pain at baseline contributed data to calculation of SEs. In the presence of significant interaction (p<0.05), predicted probabilities of reporting pain at follow-up were calculated and plotted according to Physical Activity Index Score and sleep disturbance status. Models were adjusted for age, gender, race/ethnicity, number of years of school education, BMI, self-reported history of depression, major diseases (self-reported history of cancer (any kind except skin), lung disease, heart condition or stroke), diabetes and arthritis.

## Patient involvement

Patients were not involved in this study and there are no plans to disseminate the findings to HRS participants.

## RESULTS
### Descriptive analysis

Of 16 384 HRS respondents, 16 330 (99.7%) answered the question about troublesome pain in 2014. Of these, 9828 (60.2%) reported not being troubled by pain. Baseline demographic and health-related characteristics of this pain-free sample are presented in table 1. After applying sample weights, 51.5% were female, and the mean age was 66.9.

**Table 1** Demographic and health characteristics of Health and Retirement Study respondents not troubled by pain in 2014

| | Unweighted frequency (N)* | Weighted proportion or mean (SE)* |
|---|---|---|
| **Gender** | | |
| Female | 5346 | 51.5% (0.005) |
| Male | 4482 | 48.5% (0.005) |
| Age (mean and SE) | 9828 | 66.9 (0.3) |
| **BMI** | | |
| Underweight/normal weight | 2816 | 31.2% (0.006) |
| Overweight | 3426 | 38.9% (0.007) |
| Obese | 1820 | 20.1% (0.006) |
| Obese, BMI ≥35 | 920 | 9.9% (0.004) |
| Missing data | 846 (8.6%) | |
| **Race/ethnicity** | | |
| White | 6371 | 78.7% (0.01) |
| Black | 1837 | 9.6% (0.006) |
| Hispanic | 1301 | 8.4% (0.01) |
| Other | 319 | 3.4% (0.003) |
| Years of school (mean and SE) | 9792 | 13.5 (0.08) |
| Missing data | 36 (0.4%) | |
| **Marital status in 2014** | | |
| Married | 5759 | 62.6% (0.007) |
| Separated or divorced | 1644 | 16.2% (0.005) |
| Widowed | 1943 | 14.4% (0.004) |
| Never married | 481 | 6.8% (0.004) |
| Missing data | 1 (0.01%) | |
| **Self-rated health** | | |
| Excellent | 1131 | 14.0% (0.005) |
| Very good | 3611 | 40.6% (0.008) |
| Good | 3255 | 30.7% (0.007) |
| Fair | 1499 | 12.0% (0.006) |
| Poor | 323 | 2.5% (0.002) |
| Do not know/refused | 9 (0.09%) | |
| **History of depression** | | |
| Yes | 1554 | 16.8% (0.005) |
| No | 8203 | 83.2% (0.005) |
| Missing data | 71 (0.7%) | |
| **History of major disease†** | | |
| Yes | 3766 | 35.5% (0.008) |
| No | 5960 | 64.5% (0.008) |
| Missing data | 102 (1.0%) | |
| **Alzheimer's disease or dementia** | | |
| Yes | 274 | 2.0% (0.001) |
| No | 9510 | 98.0% (0.001) |

Continued

**Table 1**  Continued

|  | Unweighted frequency (N)* | Weighted proportion or mean (SE)* |
|---|---|---|
| Missing data | 44 (0.5%) | |
| **Arthritis** | | |
| Yes | 4653 | 43.8% (0.008) |
| No | 5175 | 56.2% (0.008) |
| **Diabetes** | | |
| Yes | 2273 | 19.9% (0.005) |
| No | 7486 | 80.1% (0.005) |
| Missing data | 69 (0.7%) | |
| **Sleep disturbance** | | |
| Rarely or never | 2131 | 22.3% (0.006) |
| Sometimes | 4404 | 45.9% (0.007) |
| Most of the time | 3253 | 31.8% (0.005) |
| Missing data | 40 (0.4%) | |
| **Physical activity** | | |
| Moderate or vigorous > once a week | 5668 | 62.1% (0.008) |
| Moderate or vigorous ≤1 week | 4151 | 37.9% (0.008 |
| Missing data | 9 (0.09%) | |
| Physical Activity Index Score | 9763 | 9.0 (0.12) |
| Missing data | 65 (0.7%) | |

*Unweighted frequencies represent sample counts and weighted proportion or mean (linearised SE) represents corresponding estimated population values (through sampling weights, 9828 represent 50 923 453 community-dwelling American adults aged 50 years and above).
†Major disease defined as having a history of cancer (excluding skin), lung disease, heart condition or stroke.
BMI, body mass index.

### Demographic and health-related characteristics and incident pain

The data on reported pain status were provided by 8513 (86.6%) respondents in 2016. After applying survey weights, an estimated 18.6% (SE=0.005) of the sampled population had self-reported incident troublesome pain in 2016. This was associated with female gender, older age, higher BMI, less education, being separated, divorced or widowed, poorer self-rated health, having a history of depression, major disease, Alzheimer's disease or dementia, arthritis, diabetes, a higher frequency of sleep disturbance and lower levels of physical activity (table 2). Having missing data for pain status in 2016 was associated with older age, higher BMI, Hispanic ethnicity, less formal education, being widowed, poorer self-rated health, a history of major disease, Alzheimer's disease or dementia and lower physical activity levels in 2014.

### PHYSICAL ACTIVITY

37.9% (SE=0.008) of the baseline population participated in moderate or vigorous physical activity once a week or less. Overall, the mean baseline Physical Activity Index Score was 9.0 (SE=0.12). For those participating in moderate or vigorous physical activity once a week or less, the mean Physical Activity Index Score was 2.5 (SE=0.07); for those who reported participating in moderate or vigorous activity more than once a week, the mean Physical Activity Index Score was 12.9 (SE=0.10).

### SLEEP

22.3% (SE=0.006) of the baseline pain-free sample reported rarely or never having sleep disturbance, 45.9% (SE=0.007) reported sometimes experiencing sleep disturbance and 31.8% (SE=0.005) reported experiencing sleep disturbance most of the time.

### The association between physical activity and incident troublesome pain and the moderating effect of sleep disturbance

In models adjusted for pre-specified covariates and stratified by sleep disturbance categories, each one point higher on the Physical Activity Index Score (range=0–18) was associated with a reduced likelihood of troublesome pain onset for those who reported 'never/rarely' having sleep disturbance (OR=0.97, 95% CI 0.94 to 0.99). The protective effect attenuated with increasing frequency of sleep disturbance (sleep disturbance sometimes: OR=0.99, 95% CI 0.97 to 1.01; sleep disturbance most of the time: OR=1.01, 95% CI 0.99 to 1.03). In an adjusted model that included a Physical Activity Index Score–frequency of sleep disturbance interaction term, a significant moderating effect of sleep disturbance on the Physical Activity Index Score–incident pain association was observed (Wald test for the interaction: p=0.02) (table 3, figure 1).

### DISCUSSION

This nationally representative prospective cohort of US adults shows that self-reported sleep disturbance moderates the association between physical activity and later pain. Although higher levels of physical activity were associated with reduced likelihood of incident troublesome pain among those who reported sleep disturbance never, rarely or sometimes, the benefit of higher levels of physical activity was not observed for those who reported disturbed sleep most of the time.

Previous cross-sectional research has demonstrated linear, negative associations between increasing levels of exercise frequency, duration and intensity and the presence of chronic pain.[17] Our finding that increasing levels of physical activity are associated with a reduced likelihood of later pain reporting is consistent with this and, in addition to the prospective nature of our study, we have extended the evidence by identifying a moderating

**Table 2**  Physical Activity Index Scores in 2014 and incident pain in 2016 by demographic and health-related factors

| | 2014 Physical Activity Index | | 2016 Incident pain | |
| --- | --- | --- | --- | --- |
| | Weighted mean or B (SE) | P value | Weighted proportion or OR (SE) | P value |
| Gender | | | | |
| Female | 8.3 (0.1) | <0.001 | 20.1% (0.008) | 0.01 |
| Male | 9.6 (0.2) | | 16.9% (0.008) | |
| Age | B=−0.2 (0.009) | <0.001 | OR=1.01 (0.004) | 0.03 |
| BMI | | | | |
| Underweight/normal weight | 9.9 (0.2) | <0.001 | 15.4 (0.008) | <0.001 |
| Overweight | 9.5 (0.2) | | 16.8 (0.008) | |
| Obese | 8.4 (0.2) | | 22.5 (0.01) | |
| Obese, BMI ≥35 | 6.4 (0.2) | | 26.7 (0.02) | |
| Race/ethnicity | | | | |
| White | 9.2 (0.1) | <0.001 | 17.9 (0.006) | 0.06 |
| Black | 7.7 (0.2) | | 19.4 (0.01) | |
| Hispanic | 8.4 (0.4) | | 23.6 (0.02) | |
| Other | 8.7 (0.5) | | 19.4 (0.03) | |
| Years of school | B=0.5 (0.04) | <0.001 | OR=0.9 (0.008) | <0.001 |
| Marital status | | | | |
| Married | 9.6 (0.2) | <0.001 | 17.5% (0.007) | 0.002 |
| Separated or divorced | 9.0 (0.2) | | 21.2% (SE=0.01) | |
| Widowed | 6.7 (0.2) | | 21.6% (0.01) | |
| Never married | 7.9 (0.3) | | 16.5% (0.03) | |
| Self-rated health | | | | |
| Excellent | 12.3 (0.3) | <0.001 | 9.0% (0.01) | <0.001 |
| Very good | 9.9 (0.2) | | 14.9% (0.006) | |
| Good | 7.8 (0.1) | | 22.9% (0.01) | |
| Fair | 6.0 (0.3) | | 30.2% (0.02) | |
| Poor | 2.6 (0.2) | | 34.7% (0.03) | |
| History of depression | | | | |
| Yes | 7.9 (0.3) | <0.001 | 29.3% (0.01) | <0.001 |
| No | 9.2 (0.1) | | 16.3% (0.006) | |
| History of major disease* | | | | |
| Yes | 7.5 (0.1) | <0.001 | 22.8% (0.009) | <0.001 |
| No | 9.8 (0.2) | | 16.4% (0.007) | |
| Alzheimer's disease or dementia | | | | |
| Yes | 4.1 (0.5) | <0.001 | 30.7% (0.03) | <0.001 |
| No | 9.1 (0.1) | | 18.4% (0.005) | |
| Arthritis | | | | |
| Yes | 8.2 (0.1) | <0.001 | 26.8% (0.007) | <0.001 |
| No | 9.5 (0.2) | | 12.3% (0.006) | |
| Diabetes | | | | |
| Yes | 7.1 (0.2) | <0.001 | 23.1% (0.01) | <0.001 |
| No | 9.4 (0.1) | | 17.5% (0.005) | |
| Sleep disturbance | | | | |
| Rarely or never | 9.9 (0.2) | <0.001 | 11.8% (0.01) | <0.001 |
| Sometimes | 9.0 (0.2) | | 17.0% (0.007) | |
| Most of the time | 8.3 (0.2) | | 25.7% (0.01) | |
| 2014 Physical Activity Index | | | | |
| Moderate or vigorous >1 week | | | 17.2% (0.006) | <0.001 |
| Moderate or vigorous ≤1 week | | | 21.0% (0.008) | |
| Physical Activity Index Score | | | OR=0.97 (0.004) | <0.001 |

*Major disease defined as having a history of cancer (excluding skin), lung disease, heart condition or stroke.
B, beta; BMI, body mass index.

**Table 3** Logistic regression analysis investigating the moderating effect of sleep disturbance category on the association between 2014 Physical Activity Index Score and likelihood of troublesome pain in 2016*

| | OR (95% CI) | Linearised SE | T | P |
|---|---|---|---|---|
| Physical Activity Index Score | 0.97 (0.94 to 0.99) | 0.01 | −2.46 | 0.02 |
| Sleep disturbance category | | | | |
| Rarely or never | Reference | | | |
| Sometimes | 1.14 (0.78 to 1.67) | 0.22 | 0.68 | 0.50 |
| Most of the time | 1.42 (0.99 to 2.05) | 0.26 | 1.92 | 0.06 |
| Physical Activity Index Score–sleep disturbance category interaction term* | | | | |
| Rarely or never | Reference | | | |
| Sometimes | 1.02 (0.99 to 1.06) | 0.02 | 1.18 | 0.24† |
| Most of the time | 1.05 (1.01 to 1.08) | 0.02 | 2.79 | 0.01† |
| Age | 0.99 (0.99 to 1.00) | 0.004 | −1.57 | 0.12 |
| Gender | | | | |
| Male | Reference | | | |
| Female | 1.02 (0.85 to 1.23) | 0.10 | 0.20 | 0.84 |
| BMI | | | | |
| Underweight/normal weight | Reference | | | |
| Overweight | 1.05 (0.88 to 1.26) | 0.09 | 0.57 | 0.57 |
| Obese | 1.36 (1.09 to 1.70) | 0.15 | 2.74 | 0.01 |
| Obese, BMI ≥35 | 1.56 (1.22 to 1.99) | 0.19 | 3.63 | 0.001 |
| Race/ethnicity | | | | |
| White | Reference | | | |
| Black | 1.02 (0.85 to 1.22) | 0.09 | 0.22 | 0.83 |
| Hispanic | 1.27 (0.92 to 1.77) | 0.21 | 1.49 | 0.14 |
| Other | 1.28 (0.86 to 1.89) | 0.25 | 1.25 | 0.22 |
| Years of school | 0.95 (0.93 to 0.97) | 0.01 | −5.13 | <0.001 |
| History of depression | 1.69 (1.38 to 2.07) | 0.17 | 5.17 | <0.001 |
| History of major disease‡ | 1.24 (1.02 to 1.51) | 0.12 | 2.22 | 0.03 |
| Arthritis | 2.35 (2.00 to 2.78) | 0.19 | 10.41 | <0.001 |
| Diabetes | 1.05 (0.92 to 1.21) | 0.07 | 0.73 | 0.47 |

*Adjusted analysis: N=8036.
†Wald test statistic for overall interaction: p=0.02.
‡Major disease defined as having a history of cancer (excluding skin), lung disease, heart condition or stroke.
.BMI, body mass index.

effect of sleep disturbance on this association. Given our results, it is possible that the beneficial influence of physical activity on pain reporting has previously been underestimated, and perhaps muted by a proportion having unmeasured or unaccounted for chronic sleep disturbance.

Mechanisms by which sleep disturbance may impact the association between level of physical activity and later pain onset are likely multidimensional,[41] potentially acting as both moderators and mediators. Sleep disturbance, including delayed sleep onset, waking up during the night and waking up earlier than intended, can be associated with reduced total sleep time and, consequently, less time spent in non-rapid eye movement slow-wave sleep (SWS).[42] Adequate SWS is thought to contribute to the recuperative function of sleep. A reduced duration of SWS and, therefore, protracted restorative sleep

have been shown to reduce next-day performance[43] and may diminish the beneficial effects of physical activity by limiting recovery. Suboptimal sleep also has a negative impact on immune function,[44 45] and a body of evidence supports a case for sleep as essential to recovery after exercise in elite athletes.[46–48] It is plausible that similar mechanisms may be at play for older adults who participate in higher levels of physical activity—within the context of disturbed sleep, the benefit of higher levels of physical activity may not only be weakened, but may render sleep-disturbed individuals more susceptible to poor health outcomes.

Focusing exclusively on a working population, Skarpsno and colleagues examined the role of both occupational and leisure time physical activity on later sleep disturbance.[49] For those with pain, higher levels of objectively measured physical activity (both at work and during

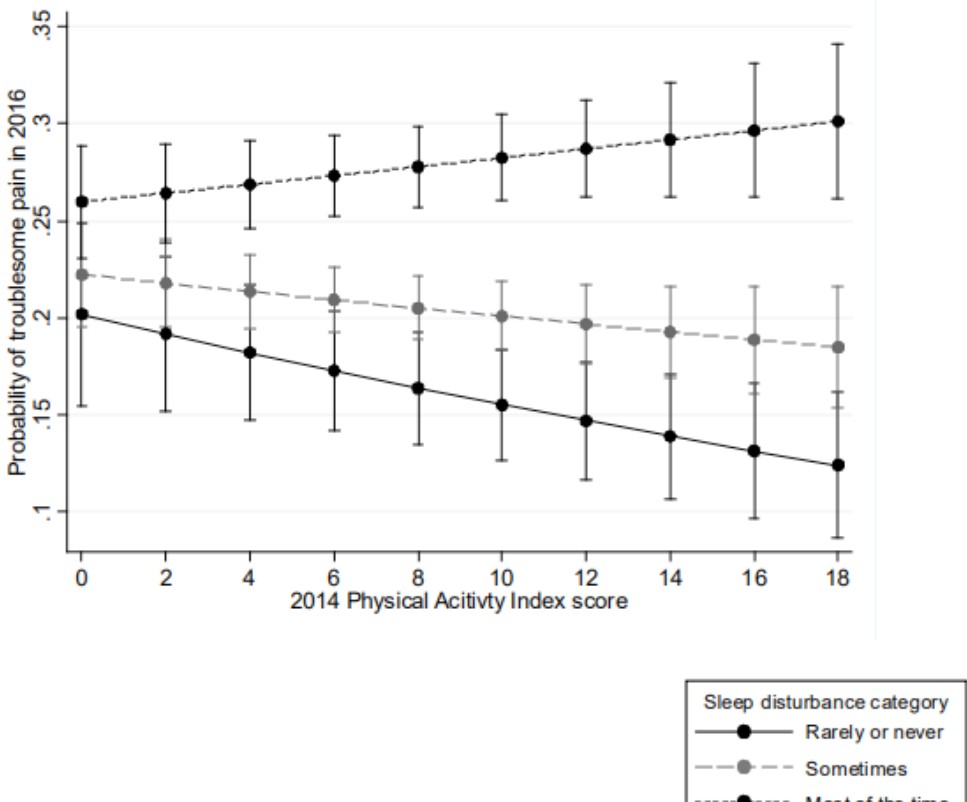

**Figure 1** The moderating effect of sleep disturbance on the association between physical activity and later incident pain.

leisure time) were associated with a higher prevalence of sleep disturbance. Skarpsno and colleagues argue that higher levels of occupational physical activity may lead to an overload response[49] (as described by Sluiter and colleagues,[50]) increasing the probability of sleep disturbance. We hypothesise that such an overload response may also be pertinent to our observation of the negative influence of sleep disturbance on the physical activity–pain relationship. In the absence of adequate restorative sleep, benefits of physical activity may be counterpointed by an aggregating overload response. However, with a population aged 50 years and above, the motivation driving different modes of physical activity may be salient; retirees who choose to participate in higher levels of physical activity may have very different experiences to those over 50 in employment, particularly those in physically demanding jobs.

We have investigated any sleep disturbance as a moderator of the physical activity–pain onset association. However, given that each individual sleep item could be argued as representing an independent sleep disturbance domain with different implications on pain reporting, in post-hoc analysis, we explored the potentially moderating effect of each specific sleep disturbance (trouble falling asleep, staying asleep, waking too early, and feeling unrefreshed on awakening). The moderating effect was most pronounced for trouble falling asleep, and all sleep disturbance domains exhibited a similar pattern of effect, with the exception that waking too early showed no sign

of moderating the physical activity–pain onset association (online Supplementary file 1). Focused research into the moderating effect of specific sleep domains, including those not available in the HRS dataset (eg, sleep duration), are recommended in future research, where more granular data collection, including objective assessments, could help to comprehensively address questions about the impact of specific sleep domains on the physical activity–pain onset association.

### Strengths and limitations
Study strengths are the relatively large sample size, representativeness of the data to the American adult population ≥50 years of age and low attrition rates. The advantages of access to and use of this dataset, however, need to be balanced against unavoidable limitations. Sleep disturbance, physical activity and pain were all assessed by self-report and are, therefore, susceptible to recall bias. Given the broad nature of response categories for pain and sleep disturbances, measurement imprecision may have resulted in misclassification that could have biased the effect estimates. The HRS questionnaire uses different response categories to the Jenkins Sleep Scale and, in the absence of comparative data, the validity of these items remains open to question. Future validation study of HRS sleep item responses against the Jenkins Sleep Scale would benefit future sleep-related research that uses the HRS dataset. Our pain outcome variable specifically is broad brush (any troublesome pain, regardless of

intensity or duration). However, the use of this outcome variable was based on previous research that suggests it is a good indicator of pain that interferes with daily activities.[35] Furthermore, it has been shown that pain is often reported as mild by people with chronic pain[51 52] and we would, therefore, have captured these cases using this single-item question.

Residual confounding cannot be excluded. We specifically chose not to take into account the consumption of medications for either sleep disturbance or pain in our analyses. From a pragmatic point of view, we were interested in problematic sleep and pain experiences—if they were being successfully managed, we considered them not to be a current problem. Also, although we adjusted for a history of depression, we did not have information on whether participants were taking antidepressant medications. Given potential effects on both sleep and pain reporting, collecting this information and including it in future analyses is recommended. An additional source of potential confounding given the age of the cohort is the built environment and living arrangements. In a post-hoc sensitivity analysis, we adjusted for nursing home status and living arrangements and the interpretation of our findings remained unchanged (online supplementary file 2). Finally, although prospective in design, this study only used two assessment time points. To provide greater insights into temporal processes, more intensive data collection and analysis would add value. Despite these limitations, our findings are consistent with existing evidence and, we believe, provide support for an argument to spend the time and finances required by further investigation.

Use of objective measures of sleep and physical activity that adhere to emerging guidelines[53] is recommended. Investigation of the possible differential impact of type of physical activity and exercise (eg, leisure time or occupational physical activity; aerobic or strengthening exercise) may also prove illuminating. It may also be worthwhile to extend these investigations to clinical populations with pain (eg, musculoskeletal, cancer and multiple sclerosis) to examine whether the sleep–physical activity interaction is also pertinent to the persistence or worsening of pain intensity or interference.

## CONCLUSION

Sleep disturbance moderated the association between physical activity and incident troublesome pain in American adults aged 50 years and above. These findings warrant further examination. Replication would provide a strong case for integrating sleep health alongside physical activity recommendations for older adults to promote healthy ageing and prevention of pain onset.

**Author affiliations**
[1]Epidemiology Group, School of Medicine, Medical Sciences and Nutrition, University of Aberdeen, Aberdeen, Scotland, United Kingdom
[2]Department of Physical Medicine and Rehabilitation, University of Michigan, Ann Arbor, Michigan, United States
[3]Department of Anesthesiology, Chronic Pain and Fatigue Research Center, University of Michigan, Ann Arbor, Michigan, United States
[4]Survey Research Center, Institute for Social Research, University of Michigan, Ann Arbor, Michigan, United States
[5]RTI International, North Carolina, United States
[6]Department of Psychiatry, University of Michigan, Ann Arbor, Michigan, United States
[7]Department of Neurology, Division of Multiple Sclerosis and Neuroimmunology, University of Michigan, Ann Arbor, Michigan, United States
[8]Department of Neurology, Division of Sleep Medicine, University of Michigan, Ann Arbor, Michigan, United States

**Acknowledgements** We gratefully acknowledge Health and Retirement Study participants who contributed data for this study.

**Contributors** DW, HG, LMS, TJB, AK and GLD were involved in study conception and design, and advised on the statistical analysis plan and interpretation of the data. HG compiled the dataset and DW performed the statistical analysis. DW drafted the manuscript. HG, LMS, TJB, AK and GLD reviewed the manuscript, provided amendments and approved the final version. DW, HG and GLD had full access to study data and take responsibility for its accuracy and the integrity of the analysis.

**Funding** DW is supported by a Foundation Fellowship Versus Arthritis (Award Number: 21742).

**Competing interests** None declared.

**Patient and public involvement** Patients and/or the public were not involved in the design, or conduct, or reporting, or dissemination plans of this research.

**Patient consent for publication** Not required.

**Ethics approval** All Health and Retirement Study data collection protocols are approved by the University of Michigan Institutional Review Board.

**Provenance and peer review** Not commissioned; externally peer reviewed.

**Data availability statement** Data are available in a public, open access repository. The dataset used for this study was generated from data products publicly released by the Health and Retirement Study (HRS): https://hrs.isr.umich.edu. The HRS is sponsored by the National Institute on Ageing (grant number NIA U01AG009740).

**ORCID iD**
Daniel Whibley http://orcid.org/0000-0002-7131-7158

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
