## [Reviewer comments · BMJ Open]

ARTICLE DETAILS

TITLE (PROVISIONAL)	Sleep disturbance as a moderator of the association between physical activity and later pain onset among American adults aged 50 and over: evidence from the Health and Retirement Study
AUTHORS	Whibley, Daniel; Guyer, Heidi; Swanson, Leslie M.; Braley, Tiffany J.; Kratz, A; Dunietz, Galit Levi

VERSION 1 – REVIEW

REVIEWER	Simon Hayhoe Colchester University Hospital, England
REVIEW RETURNED	20-Jan-2020

GENERAL COMMENTS	This is a useful addition to the literature derived from the ongoing American Health and Retirement Study. The two-way relationships between sleep and pain, between exercise and pain, and between sleep and exercise are all well known. This paper explores the three-way relationship and discovers a more complicated and unexpected association, where poor sleep not merely reduces the benefits of exercise, but actually increases the likelihood of pain for high levels of exercise in the elderly. This is an important finding. The paper is well and logically presented, although the statistical analyses are beyond my capabilities. I found the “Demographic and health ...” section of the Results confusing. It is not clear what “population” is being referred to in the first sentence. If it refers to the 8,513, then I would suggest changing the sentence to “Data on pain status was provided by 8,513 (86.6%) respondents in 2016: an estimated 18.6% (SE 0.005) of these reported ...” (Why is this figure estimated?). However, the figure 8,036 (82%) responses available for analysis given in the abstract differs from the 8,513 above, and does not appear here in the results section. I assume that the only follow-up measure recorded from the 2016 data was that of troublesome pain, and that changes in marital or medical status were not taken into consideration. So why is there a discrepancy of 477 analysed? In the Discussion section reference 49 is not noted in the text, and the sentence including reference 51 should start “Sluiter and colleagues”, not Skarpsno.
--

REVIEWER	Ninad Chaudhary The University of Alabama at Birmingham, US
REVIEW RETURNED	27-Feb-2020

GENERAL COMMENTS	The authors have presented an important topic on understanding the relationship between physical activity and pain in the elderly
---

population and subsequent role of sleep disturbance in this relationship. The authors found that the association between physical activity index score and incident troublesome pain is moderated by sleep in the elderly population such that the probability of having pain in 2016 decreases with improved physical activity index score among those who rarely report sleep disturbances. The reviewer agrees with the authors that the research efforts of incorporating sleep behavior in physical activity guidelines for older adults is limited and should be strongly pursued.

Major Concerns:

1. The original Jenkins Sleep Scale (Ref 39) scored items on the point scale of 0-5 depending on the number of days participants experience symptoms. However, the sleep scale here seems to be modified into three categories (rarely or never; sometimes; most of the time). How do these new categories stand in comparison to the original category? Has the modified scale categories validated elsewhere?
2. From the perspective of the outcome – pain, did the authors think of looking at each sleep question as a moderator separately? Each sleep question is an independent endo-phenotype and may have differential implications on the pain. PMID: 30314881 reports that those with chronic pain more likely to have problems with sleep initiation or sleep maintenance. While these effects could be down-stream to chronic pain, it will be interesting to examine their role in affecting the physical activity and pain onset.
3. Table 3. Is the Odds ratio for physical activity represents the adjusted odds ratio after including interaction term or without including the interaction term? Assuming they represent with interaction term in the model, the odds ratio is pretty similar to the “never/rarely” group. In comparison to other sub-group odds ratio, the main association is pretty similar. It is of concern that the sleep disturbance can indeed be considered as a moderator/effect modifier based on these small changes in the effect sizes and interaction p-values. These associations could be a function of different N in each sub-group. Did the authors consider that sleep disturbances that we are observing are non-collapsibility of odds ratio with no effect modification? What are the effect estimates for the association between physical activity and sleep disturbance variables?
4. Additionally, both the definitions of pain and sleep disturbance being broader, the small change in estimates could be biased due to misclassification/measurement error.
5. Considering this is a slightly elderly population, have the authors considered additional factors that could affect physical activity such as built-environment, a place to stay (assisted facilities, nursing home, residence) which could be equally important effect modifiers/confounders in the relationship? The lack of these factors seem to be an important limitation here.

Minor Concerns:

Introduction:

1. “although changes are observed..... “ suggests that lower sleep duration indicates sleep disturbance which is not true

Methods:

2. Page 7 Line 12 Reference for WHO recommendations is missing. It will also be important to consider whether such guidelines can have regional variation and may not be truly applied to the population in Michigan.

	3. The authors should explicitly state here the study cycle when sleep disturbances were measured. 4. For the history of depression – did the authors have information on depressive medications? It will be an important measure to consider as these medications have some effect both on sleep and pain sensitivity. 5. Were those with missing pain status were statically different from the population under study? Discussion: 6. Page 17- Line 52 --- it is a slight overstatement to consider that age can be a proxy for employment status. Also, Skarpsno’s paper is focused on the higher activity at occupation accounting for the type of occupation-based physical activity. With no information on the type of occupation, the discussion here seems quite hypothetical. 7. Page 17 Line 12 – the authors built up the rationale where sleep seems to be the mediator of the relationship between physical activity and poor health outcomes rather than a moderator.
--	---

VERSION 1 – AUTHOR RESPONSE

Reviewer: 1

This is a useful addition to the literature derived from the ongoing American Health and Retirement Study. The two-way relationships between sleep and pain, between exercise and pain, and between sleep and exercise are all well known. This paper explores the three-way relationship and discovers a more complicated and unexpected association, where poor sleep not merely reduces the benefits of exercise, but actually increases the likelihood of pain for high levels of exercise in the elderly. This is an important finding. The paper is well and logically presented, although the statistical analyses are beyond my capabilities.

1. I found the “Demographic and health ...” section of the Results confusing. It is not clear what “population” is being referred to in the first sentence. If it refers to the 8,513, then I would suggest changing the sentence to “Data on pain status was provided by 8,513 (86.6%) respondents in 2016: an estimated 18.6% (SE 0.005) of these reported ...” (Why is this figure estimated?).

Dear Reviewer,

Thank you for providing comments and suggestions on our manuscript. The changes we refer to can be found on the marked-up document.

You are correct – we are referring to the 8,513 respondents who had no pain at baseline and provided data on the outcome (pain status) at follow-up (in 2016). To make this clearer we have changed the wording as you have suggested. Please see page 9 of the marked up document:

“Data on reported pain status was provided by 8,513 (86.6%) respondents in 2016.”

With regard to the use of the term ‘estimated’, the proportion is an estimate of the US population aged 50 years and over after applying the sample weights that are included in the complex survey design of the Health and Retirement Study. To make this clearer we have made the following change on page 9 of the marked-up document:

“After applying survey weights, an estimated 18.6% (SE 0.005) of the sampled population had self-reported incident troublesome pain in 2016.”

2. However, the figure 8,036 (82%) responses available for analysis given in the abstract differs from the 8,513 above, and does not appear here in the results section. I assume that the only follow-up measure recorded from the 2016 data was that of troublesome pain, and that changes in marital or medical status were not taken into consideration. So why is there a discrepancy of 477 analysed?

8,513 respondents provided information about pain status in 2016. However, as described at the top of page 8, under ‘Demographic and Health Characteristics’, we adjusted for pre-specified covariates (potential confounders) in our models (age, gender, race/ethnicity, and number of years of school education, Body Mass Index, history of depression, major disease, diabetes and arthritis). A complete case analysis meant that some participants with pain data in 2016 were not retained in adjusted analysis, leaving an adjusted sample size of 8,036. This information is included in the abstract:

“Of 9828 eligible baseline respondents, 8036 (82%) had complete follow-up data for adjusted analyses”

To address your comment and make this more explicit in the results, we have included a footnote to Table 3 on page 15 that states the adjusted analysis sample size:

“* Adjusted analysis N=8,036”

3. In the Discussion section reference 49 is not noted in the text, and the sentence including reference 51 should start “Sluiter and colleagues”, not Skarpsno.

Thank you for identifying the error with the referencing, We have gone through and checked all referencing thoroughly and have made all necessary changes.

Major Concerns:

1. The original Jenkins Sleep Scale (Ref 39) scored items on the point scale of 0-5 depending on the number of days participants experience symptoms. However, the sleep scale here seems to be modified into three categories (rarely or never; sometimes; most of the time). How do these new categories stand in comparison to the original category? Has the modified scale categories validated elsewhere?

Dear Reviewer,

Thank you for providing comments and suggestions on our manuscript. The changes we refer to can be found on the marked-up document.

The Health and Retirement Study questionnaire only includes the response categories for the sleep disturbance questions that we have used in our study (rarely or never; sometimes; most of the time). We are therefore unable to compare this categorization with the original Jenkins Sleep Scale. Although these items from the Health and Retirement Study have been used in previous studies as a measure of sleep disturbance (e.g. DOI 10.1007/s10865-015-9635-4), we are unaware of existing validation studies that have compared them with the Jenkins Scale. We have highlighted this in the discussion and have recommend validation of the Health and Retirement Study items in a future study. In response to your feedback, we have included this as a limitation in the discussion section of the manuscript:

p18: "The HRS questionnaire uses different response categories to the Jenkins Sleep Scale and, in the absence of comparative data, the validity of these items remains open to question. Future validation study of HRS sleep item responses against the Jenkins Sleep Scale would benefit future sleep-related research that uses the HRS dataset."

2. From the perspective of the outcome – pain, did the authors think of looking at each sleep question as a moderator separately? Each sleep question is an independent endo-phenotype and may have differential implications on the pain. PMID: 30314881 reports that those with chronic pain more likely to have problems with sleep initiation or sleep maintenance. While these effects could be down-stream to chronic pain, it will be interesting to examine their role in affecting the physical activity and pain onset.

We agree that there is merit in examining potentially moderating effects of each specific sleep domain available in the Health and Retirement study dataset and have undertaken this analysis in response to your comment. A description of this additional post-hoc analysis has been added to the discussion section of the manuscript on page 18 of the marked up document:

"We have investigated any sleep disturbance as a moderator of the physical activity-pain onset association. However, given that each individual sleep item could be argued as representing an

independent sleep disturbance domain with different implications on pain reporting, in post-hoc analysis we explored the potentially moderating effect of each specific sleep disturbance (trouble falling asleep, staying asleep, waking too early, and feeling unrefreshed upon awakening). The moderating effect was most pronounced for trouble falling asleep, and all sleep disturbance domains exhibited a similar pattern of effect, with the exception that waking too early showed no sign of moderating the physical activity-pain onset association. Focused research into the moderating effect of specific sleep domains, including those not available in the HRS dataset (e.g. sleep duration) are recommended in future research where more granular data collection, including objective assessments, could help to comprehensively address questions about the impact of specific sleep domains on the physical activity-pain onset association.”

3. Table 3. Is the Odds ratio for physical activity represents the adjusted odds ratio after including interaction term or without including the interaction term? Assuming they represent with interaction term in the model, the odds ratio is pretty similar to the “never/rarely” group. In comparison to other sub-group odds ratio, the main association is pretty similar. It is of concern that the sleep disturbance can indeed be considered as a moderator/effect modifier based on these small changes in the effect sizes and interaction p-values. These associations could be a function of different N in each sub-group. Did the authors consider that sleep disturbances that we are observing are non-collapsibility of odds ratio with no effect modification? What are the effect estimates for the association between physical activity and sleep disturbance variables?

The adjusted ‘main effect’ of physical activity level on odds of later pain onset, stratified by sleep disturbance category, is provided in text on page 10. These odds ratios represent the effect of an increase of 1 point on the continuous physical activity scale on likelihood of future pain reporting. The overall effect accumulates with each additional increase in physical activity index score.

Table 3 presents all coefficients for the model that includes the interaction term. As the interaction is statistically significant, we do not interpret the ‘main effects’ from this model. We have used output from this ‘interaction’ model to calculate predicted probabilities of pain onset and have plotted these according to physical activity index score and sleep disturbance status (Figure 1). This figure depicts the effect across the full range of physical activity levels. A discernable difference in the slopes according to sleep disturbance status can be observed.

In Table 2 we provide data for the weighted mean level of baseline physical activity for each sleep disturbance category, and have reported statistically significant differences such that, on average, activity levels are lower when sleep problems are more frequent. This is, though, only ‘on average’ and a range of physical activity levels are observed in each sleep disturbance category.

With regard to concerns about non-collapsibility of odds ratios, our research question and analyses have been guided by subject matter knowledge. Our primary variables of interest are assessed as 1 continuous and 1 categorical variable (physical activity index and pain onset, respectively) and 1 ordinal variable examined as a putative moderator (sleep disturbances). Our results are not

surprising and, given that the associations are qualitatively similar in crude analysis and after partial and 'full' adjustment, we do not believe that a 'Simpson's paradox' effect is present.

4. Additionally, both the definitions of pain and sleep disturbance being broader, the small change in estimates could be biased due to misclassification/measurement error.

We do not disagree with this point and have included this as a limitation of our study (p.19):

"Given the broad nature of response categories for pain and sleep disturbances, measurement imprecision may have resulted in misclassification that could have biased the effect estimates."

However, given the broad definitions of pain onset and sleep disturbance that we have used, we think that misclassification and measurement error are very unlikely for those with definite sleep or pain problems (misclassification in this case would be most likely for those close to the threshold of the admittedly broad categories). We agree that more granular data would help to achieve a more precise answer to our research question. It's our hope that our study, as a starting point, may provoke the collection and analysis of more granular data in the future.

5. Considering this is a slightly elderly population, have the authors considered additional factors that could affect physical activity such as built-environment, a place to stay (assisted facilities, nursing home, residence) which could be equally important effect modifiers/confounders in the relationship? The lack of these factors seem to be an important limitation here.

We have returned to the source dataset and in a sensitivity analysis have made adjustments for nursing home status and living arrangements. After making these adjustment, the regression coefficients in our models changed only very slightly and their interpretations remained unchanged. We have added this information to the discussion section of the manuscript:

Bottom of p19 of the marked-up document: "An additional source of potential confounding given the age of the cohort is the built environment and living arrangements. In a post-hoc sensitivity analysis we adjusted for nursing home status and living arrangements and the interpretation of our findings remained unchanged."

Minor Concerns:

Introduction:

- 1. "although changes are observed..... " suggests that lower sleep duration indicates sleep disturbance which is not true**

In this section of the manuscript we state that there may be a tacit societal agreement that different parameters of sleep, such as duration, timing and continuity decline with age. We do not relate this to sleep disturbance. Our statement that 'although changes are observed as humans age' relates to

observed changes in parameters of sleep like duration, timing and continuity. Again, we do not relate this to sleep disturbances here. To respond to your comment and to ensure that our meaning is clear, we have made the following change to the text on p. 5

“However, although changes in sleep duration, timing and continuity are observed as humans age...”

Methods:

2. Page 7 Line 12 Reference for WHO recommendations is missing. It will also be important to consider whether such guidelines can have regional variation and may not be truly applied to the population in Michigan.

The reference for WHO physical activity recommendation is included at the start of this paragraph on physical activity. We therefore think it is superfluous to reiterate the reference within the same paragraph:

p.5, Section titled ‘Physical activity’: “The World Health Organization (WHO) recommends that adults aged ≥ 65 years participate in 150 minutes of moderate physical activity or 75 minutes of vigorous physical activity each week, or an equivalent combination of both³⁶.”

The population under study is not restricted to Michigan. The Health and Retirement Study is conducted out of the University of Michigan but collects data from participants across the contiguous United States. To make this clearer to the reader we have added this detail to page 5:

“We used data from the Health and Retirement Study (HRS), a prospective survey of US adults aged 50 years and older conducted by the University of Michigan. Household face-to-face or telephone interviews are conducted with participants on a biannual basis across the contiguous United States³⁴.”

3. The authors should explicitly state here the study cycle when sleep disturbances were measured.

Thank you for pointing out this omission. We have added the following text to the methods to address this (p. 7):

“Sleep disturbances assessed at baseline (2014) were investigated as a potential moderator of the physical activity (2014)-later pain reporting (2016) association.”

4. For the history of depression – did the authors have information on depressive medications? It will be an important measure to consider as these medications have some effect both on sleep and pain sensitivity.

Unfortunately, we do not have information on medication for depression. We agree this may be a source of residual confounding and have included the following text to the manuscript to make this clear:

p.19: “Also, although we adjusted for a history of depression, we did not have information on whether participants were taking antidepressant medications. Given potential effects on both sleep and pain reporting, collecting this information and including it in future analyses is recommended”

5. Were those with missing pain status were statically different from the population under study?

This information is provided on page 9:

“Having missing data for pain status in 2016 was associated with older age, higher BMI, Hispanic ethnicity, less formal education, being widowed, poorer self-rated health, a history of major disease, Alzheimer’s disease or dementia, and lower physical activity levels in 2014.”

Discussion:

6. Page 17- Line 52 --- it is a slight overstatement to consider that age can be a proxy for employment status. Also, Skarpsno’s paper is focused on the higher activity at occupation accounting for the type of occupation-based physical activity. With no information on the type of occupation, the discussion here seems quite hypothetical.

In response to your comment we have removed the overstatement that age might be a proxy for employment status (bottom of page 17). We have, however, retained a description of Skarpsno’s findings as we believe that, although related to occupational activity, the overload response may be applicable to our population and findings. To ensure that the reader is aware that our argument is hypothetical we have made the following change:

p. 17: “We hypothesize that such an overload response may also be pertinent to our observation of the negative influence of sleep disturbance on the physical activity-pain relationship.”

7. Page 17 Line 12 – the authors built up the rationale where sleep seems to be the mediator of the relationship between physical activity and poor health outcomes rather than a moderator.

We agree that our argument here covers the potential for both moderating and mediating effects of sleep on the physical activity-pain association. Indeed, we believe that both moderating and mediating effects deserve further research attention in this context. We have made the following amendment to the text in light of your comment and our broader research agenda:

p.16

“Mechanisms by which sleep disturbance may impact the association between level of physical activity and later pain onset are likely multidimensional⁴¹, potentially acting as both moderators and mediators.”

VERSION 2 – REVIEW

REVIEWER	Simon Hayhoe Essex Partnership University Trust, UK
REVIEW RETURNED	17-Apr-2020

GENERAL COMMENTS	Thank you: my comments have all been dealt with effectively in this revised version.
--

REVIEWER	Ninad Chaudhary UAB
REVIEW RETURNED	30-Apr-2020

GENERAL COMMENTS	The authors have provided a satisfactory and well-concerted response to the comments. I have only one additional comment to consider: Comment 2 and Comment 5 Response: I appreciate the author's response on looking at each sleep item independently as well as considering built-environment factors in their sensitivity analysis. The authors highlighted their findings in the discussion section. However, it seems that the authors have not added them as additional results to the manuscript or provided them in the supplementary material to support their statements. I would leave it to the editor's discretion whether that is ok with the journal format.
--

VERSION 2 – AUTHOR RESPONSE

Reviewer 2: We have now included the sensitivity analyses described in the discussion as supplementary data, as described above.